# Time-restricted eating in early-stage Huntington's disease: A 12-week interventional clinical trial protocol

**Russell G. Wells**[ID][1]*, **Lee E. Neilson**[ID][1,2], **Andrew W. McHill**[3], **Amie L. Hiller**[1,2]

1 Department of Neurology, Oregon Health and Science University, Portland, Oregon, United States of America, 2 Neurology and PADRECC VA Portland Health Care System, Portland, Oregon, United States of America, 3 Sleep, Chronobiology, and Health Laboratory, School of Nursing, Oregon Health and Science University, Portland, Oregon, United States of America

* wellsru@ohsu.edu

## Abstract

Huntington's disease (HD) is a devastating neurodegenerative disorder characterized by a variety of debilitating symptoms including abnormal motor control, cognitive impairment, and psychiatric disturbances. Despite significant efforts, efficacious treatments to alter the course of HD remain elusive, highlighting the need to explore new therapeutic strategies, including lifestyle changes that may delay the onset of symptoms and slow disease progression. Recent research indicates that time-restricted eating (TRE), a type of intermittent fasting where caloric intake is confined to a specific time window each day, may be beneficial in treating neurodegenerative diseases like HD. TRE has been found to enhance mitochondrial function, stimulate autophagy, lower oxidative stress, and improve cognitive performance. Although TRE has shown potential in HD animal models and non-HD populations, it has yet to be analyzed for safety, feasibility, and efficacy in persons with HD. Therefore, we propose a prospective interventional, open-label, single-arm, pilot study of 25 participants with late prodromal and early manifest HD to evaluate participant adherence to TRE diet – specifically, maintaining a 6-8-hour eating window every day for 12 weeks. Secondary measures will include pre- versus post-intervention assessment of body composition via bioelectrical impedance analysis, vital signs and safety labs, serum biomarkers of neurodegeneration, and standard HD behavioral, cognitive, and motor function clinical scales. Additional exploratory measures will evaluate sleep quality, physical activity, mood, dietary composition, and mitochondrial function. We expect that the diet will be safe, feasible, and may also improve biomarkers of disease progression in persons with HD. We anticipate this study will lay the foundation for future large-scale clinical trials to further evaluate the clinical efficacy of TRE in HD. This study has been registered on July 8, 2024 with ClinicalTrials.gov registration number NCT06490367 (https://clinicaltrials.gov/study/NCT06490367).

**Data availability statement:** No datasets were generated or analyzed during the current study. All relevant data from this study will be made available upon study completion.

**Funding:** This work is supported by a PCO pilot award (to R.G.W), VA CSR&D CDA2 IK2 CX00253-01A1 (to L.E.N.), the Parkinson's Disease Research, Education, Clinical Center at the VA Portland Health Care System, and by the National Center for Advancing Translational Sciences, National Institutes of Health, through Grant Award Number TL1TR002371 (to R.G.W) and UL1TR002369. The content is solely the responsibility of the authors and does not necessarily represent the official views of the NIH. Role of Funder statement: The funders are institutions and institutional grants. Some of the authors are employees of the institutions but there were no institutional representatives beyond the authors of the paper that were involved in the decision to publish or preparation of the manuscript. The acknowledgement section lists other institutional employees who were involved in the study design.

**Competing interests:** The authors have declared that no competing interests exist.

## Introduction

Huntington's disease (HD) is a devastating hereditary neurodegenerative disorder characterized by a triad of motor symptoms (including involuntary and bradykinetic movements), cognitive impairment, and psychiatric abnormalities that ultimately progress toward death over the course of 15 to 20 years from the time symptoms appear [1,2]. With no known disease-modifying treatments available, there exists a desperate need for the discovery of effective therapeutic strategies, including ones that can be broadly accessed by those with limited resources, such as lifestyle changes that may delay the onset of symptoms and slow disease progression. Recent research indicates time-restricted eating (TRE), a type of intermittent fasting where caloric intake is confined to a specific time window each day, may be beneficial in treating neurodegenerative diseases like HD [3–6].

HD is caused by an expanded CAG repeat that results in a mutant huntingtin protein (*mHtt*) which is responsible for inducing the neuronal and cellular dysfunction that characterize the pathophysiology of the disease [7–9]. To date, most therapeutic strategies have aimed at modifying the expression of the pathogenic huntingtin gene; however, insufficient understanding of the specific functions of wild-type *Htt*, combined with the varied harmful effects caused by *mHtt* has made targeting the underlying pathology and ameliorating its phenotypic expression difficult. In addition, while a well-known association between CAG repeat length and age of onset exists, there is still large variability observed in symptom onset, severity, and progression, which suggests other mediators are involved [10,11]. Genome-wide association studies have unveiled other genes, DNA repair mechanisms, and mitochondrial redox factors as contributors to this variability [12,13]. Additionally, as seen in other neurodegenerative disorders, environmental factors and lifestyle habits are thought to modify age of HD onset and symptom severity; however, interventional human studies are still lacking in this space [14,15]. Identifying and applying healthful lifestyle strategies could have therapeutic potential in delaying HD progression and is of urgent need for HD patients and their families.

Contemporary evidence has revealed the intriguing potential of TRE to attenuate the progression of many neurodegenerative disease processes [3–6], with a growing body of literature suggesting TRE may be effective for use in HD [16]. In brief, the dietary intervention has been found to induce autophagy and clear *mHtt,* upregulate cytoprotective genes and the expression of brain derived neurotrophic factor (BDNF), stimulate mitochondrial biogenesis and improve bioenergetic functions, prevent oxidative stress, and regulate the circadian rhythm (Fig 1) [16]. Studies of TRE in HD transgenic animal models have found it increases *mHtt* clearance in the brain and attenuates the formation of huntingtin inclusions, motor dysfunction, glucose intolerance, and tissue wasting, all while extending animal lifespan and improving circadian rhythm synchronization [17–20]. Although there is very little experience with TRE in HD, a case study in a 41-year-old male HD patient with progressively deteriorating symptoms, reported that a 48-week combined metabolic strategy of TRE paired with a ketogenic diet resulted in improved motor symptoms, activities of daily living, composite Unified HD Rating Scale (UHDRS) score, and psychiatric symptoms [21]. In a 36-month longitudinal study consisting of 99 elderly subjects with mild cognitive impairment, consistent adherence to a TRE diet was associated with significant cognitive enhancements, decreased markers of DNA damage and inflammation, and improved markers in oxidative stress [6]. Because TRE diets have been applied as a weight management strategy in overweight individuals, safety is a primary concern as HD is known for causing weight loss, especially in later disease stages [22,23]. It is interesting and reassuring that a study evaluating the safety and tolerability of TRE in healthy midlife and older adults found that the diet had no influence on body mass, fat-free mass, bone density, or nutrient intake [24].

Although TRE has shown potential in animal studies and non-HD populations, it has yet to be analyzed for safety, feasibility, or efficacy in the clinical setting. To address this clinical relevance and lack of existing knowledge, we will perform an interventional study in persons with premanifest and early manifest HD to examine if a TRE diet appears safe and feasible (Fig 3). We will also explore efficacy regarding HD symptoms and biomarkers of disease progression. Our goal is to look at TRE in HD on a larger and longer scale. The rationale for this study is grounded in the compelling physiological mechanisms underlying TRE's therapeutic potential as a simple lifestyle change to reshape the course of HD progression. It is also critical to discourage such practices if we see negative consequences of TRE. Using what is known of the TRE mechanism of action and current data, we expect that the diet will be safe, feasible, and may also improve biomarkers of disease progression and motor, cognitive, and behavioral function in patients with manifest HD. If this is true, our study will lay the foundation for future large-scale clinical trials to prove the efficacy of TRE in a life-threatening neurodegenerative disease.

## Objectives

1. Primary: Examine the feasibility and tolerability of TRE through measures of protocol implementation, adherence rates, and adverse events.

2. Secondary: Evaluate the safety of short-term TRE in the early stages of Huntington's disease (HD) by measures of body composition, vital signs, and blood analysis.

3. Secondary: Analyze biomarker dynamics via peripheral markers of neurodegeneration and explore bioenergetic effects of TRE via measures of mitochondrial function.

4. Exploratory: Explore whether TRE has effects on behavioral, cognitive, and motor function outcomes using standard HD clinical scales.

## Methods

### Aim

The primary aim of this study is to assess the safety, feasibility, and biomarker effects of a TRE diet in persons with HD. We will also explore clinical effects and potential mechanism of action.

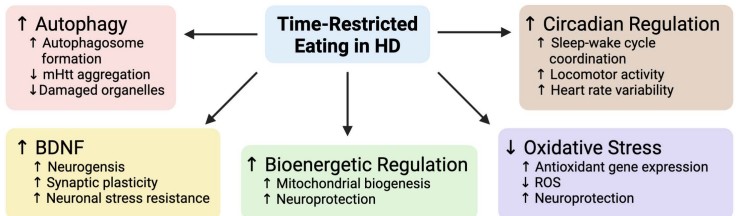

**Fig 1. Mechanisms underlying the therapeutic potential of time-restricted eating in Huntington's disease.**
Time-restricted eating (TRE), a form of intermittent fasting, in animal models of Huntington's disease (HD) and non-HD human and animal studies reveal that the practice increases autophagic activity which is thought to decrease aggregation of the mutant huntingtin protein (mHtt), stimulates production of brain derived neurotrophic factor (BDNF), improves metabolic functions, promotes oxidative stress resistance and decreases reactive oxygen species (ROS), and improves measures of circadian rhythm function. Republished from Wells et al., 2024 [16] under a CC BY license, with permission from BMC Springer Nature, original copyright 2024.

## Study design

The proposed study will be a prospective interventional, open-label, single-arm trial (Fig 2). After baseline testing and a 1-week lead-in period to assess typical eating behavior, enrolled participants will be asked to follow a TRE diet, specifically maintaining a 6-8-hour eating window every day for 12 weeks. Participants will be allowed to self-select the timing of the eating window; but once selected, they will be asked to maintain that schedule for the duration of the study. Outside of that window, for the remaining 16-18 hours of day/night, participants will be asked not to consume calorie-containing food or drink. We will measure body weight and composition, safety labs, adherence to the diet, dietary composition, sleep, physical activity, mood, and markers of efficacy. Data collection episodes will take place at the Oregon Health and Science University (OHSU) Oregon Clinical and Translational Research Institute (OCTRI) outpatient clinic within 1-3 weeks prior to the start of the study, and again within 7 days after 12 weeks of TRE. Participants will complete study surveys directly in Qualtrics™. Survey invitations and reminders will be sent by email or SMS text message to study participants. A maximum of 3 reminders will be sent to a participant for any incomplete form. Participants will be contacted via phone call if there are 2 or more days without completing meal tracking surveys.

## Inclusion criteria

Subjects eligible to participate in this study are persons who:

1. Are of at least 21 years of age at screening.

2. Must fulfill one of the following criteria:a. Premanifest late prodromal HD as defined by a genetically confirmed CAG repeat ≥ 36 and a CAG-Age Product (CAP) score > 368 (CAP = (Age) x (CAG – 33.66)) [25].

b. Early manifest (stage I and II) HD as defined by a Total Functional Capacity (TFC) greater than or equal to 7. Subjects must have been determined to have a clinical diagnosis of HD by the site investigator as defined by a diagnostic confidence level (DCL) of 4.

3. Must fulfill both of the following criteria:a.    Have undergone genetic testing with a known CAG repeat greater than or equal to 36.

b. No features of juvenile HD (Westphal variant)

Clarification of CAG Repeat Number (Allele length) Testing Requirements:
   A CAG repeat number obtained prior to the Screening Visit will be used to document subject eligibility if at Screening there is documentation available in the subject's record that states that the subject has an expanded CAG repeat (greater than or equal to 36) from a prior validated laboratory assessment.

4. All female subjects of childbearing potential must have a negative urine pregnancy test at baseline, and female subjects of childbearing potential must practice a highly effective method of contraception (e.g., oral contraceptives, a barrier method of birth control [e.g., condoms with contraceptive foams, diaphragms with contraceptive jelly], intrauterine devices, partner with vasectomy or sexual abstinence) for the duration of the study.

5. Are willing and capable of providing informed consent for study participation.

6. Are capable of reading, writing, and communicating effectively with others.

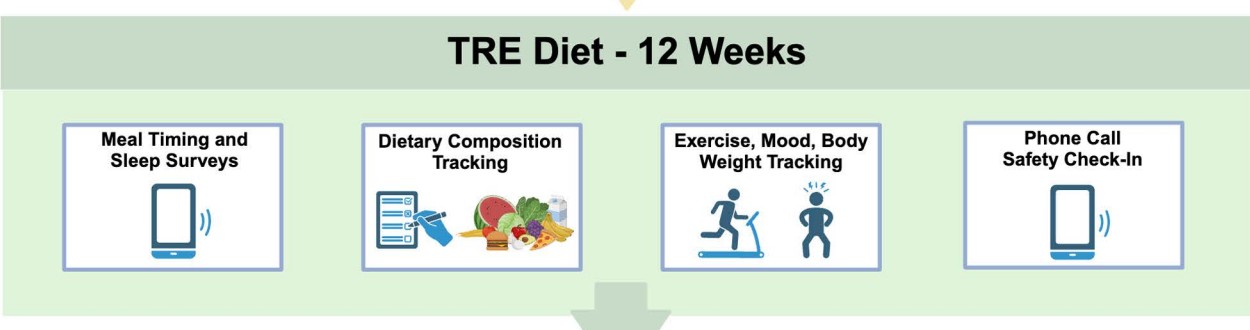

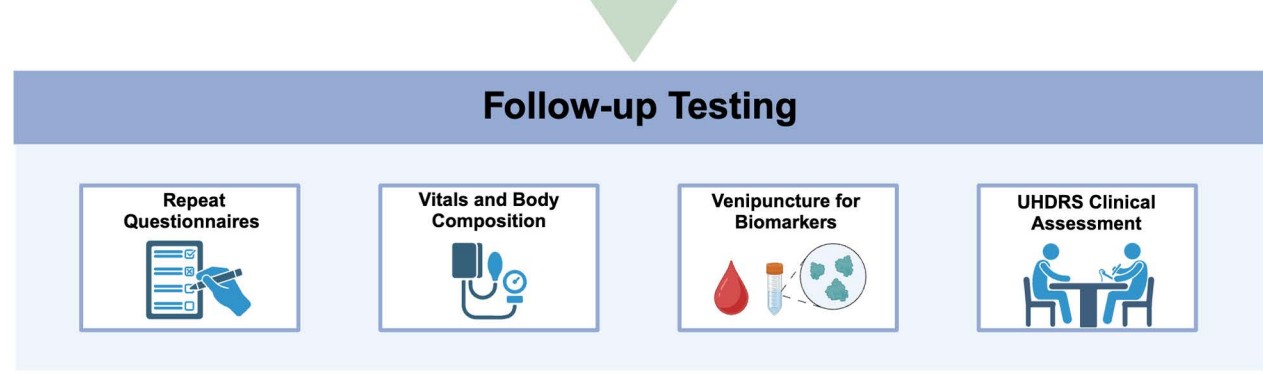

**Fig 2. Time-restricted eating in Huntington's disease study design.** We propose a prospective interventional, open-label, single-arm trial with a target enrollment of 25 participants. Baseline testing will take place approximately one week prior to the TRE onset. The lead-in period will consist of seven days of lifestyle tracking which will be followed by 12 weeks of TRE, where participants will be asked to consume all daily caloric intake within a 6-8 hour window and fast the remaining 16-18 hours of the day. They will then return to clinic for follow-up assessments. Figure created with BioRender.com.

## Exclusion criteria

Subjects ineligible to participate in this study are persons who:

1. Have participated in an investigational drug or device study within 30 days of the baseline visit

2. Have had previous neurosurgery for Huntington's disease or other movement disorders.

3. Have clinically significant cognitive impairment that hinders the ability to appropriately consent or adhere to detailed study directions, in the opinion of the principal investigator.

4. Have clinically significant psychosis and/or confusional states, in the opinion of the principal investigator.

5. Have clinically relevant hematologic, hepatic, cardiac, thyroid, or renal disease.

6. Have a history of substance abuse (based on DSMV criteria) within the past 12 months prior to screening.

7. If female, are pregnant or breastfeeding.

8. Have a high-risk for nutritional deficiency.

9. Have a change in greater than 2 kg in body mass in the preceding three months.

10. Express a desire to lose weight during the study.

11. Have a clinically significant medical (including a current or prior eating disorder), surgical, laboratory, or behavioral abnormality which in the judgment of the site Investigator makes the subject unsuitable for the study.

12. Have consistently practiced a time-restricted eating protocol within 3 months of trial onset.

## Sample size

We will recruit 40 participants expecting roughly 25 participants (62.5%) to meet inclusion criteria with the goal of yielding a final sample of at least 20 in the experimental condition after a conservative expected level of attrition (~20%). Aside from a single patient case report, an examination of TRE in persons with HD has never been explored. Thus, we must rely on existing data on TRE intervention in other populations. To detect differences in the percent change in body weight from baseline to follow-up, assuming a standard deviation of 2.4% from Gabel et al., we will have 95% power to detect a difference as low as 2.5% change in body weight in n = 15 with two-sided Type I error of 0.05 [22]. To detect differences in plasma neurofilament light protein (NfL) from baseline to follow-up, assuming a standard deviation of 0.65 log pg/ml from Byrne et al., we will have 90% power to detect a difference as low as 15% in plasma NfL between baseline and follow-up in n = 20 with two-sided Type I error of 0.05 [26]. *Thus, we will study 20 participants to maximize power while balancing feasibility* to derive meaningful data.

## Recruitment

Subjects will be recruited through the OHSU Movement Disorders clinic patient pool using a combination of approaches, including clinician recruitment and referral during clinic appointments, matching from an OHSU Institutional Review Board (IRB) approved research registry (OHSU IRB #8049), advertising via an IRB approved flyer, and contacting

participants of Enroll-HD (OHSU IRB #21855). Potential participants will be contacted either directly in the clinic or by phone, utilizing an IRB-approved phone script.

## Consent process

Potential participants will first be contacted by phone and study staff will utilize an IRB-approved phone script to assess initial interest and eligibility for the study. Those who remain interested and eligible will be provided the consent form for review (S2). The potential participant will be contacted by phone a few days later and if still interested, will be scheduled for a baseline visit. At the baseline visit, investigators will go over each aspect of the consent form and answer any questions the participants may have. When the participant is satisfied with his/her understanding of the study, and fully comprehends his/her freedom to withdraw at any time, the potential participant will be asked to sign the paper research consent form if they wish to do so.

## Procedures and data collection

One to three weeks prior to the onset of the study and after week 12 of TRE, participants will visit OHSU for the baseline and follow-up measurements (Fig 3). At the baseline visit, participants will complete the MoCA and, if female, will take a urine pregnancy test to confirm eligibility. All eligible participants will then be asked to provide written informed consent to be involved in the study. They will then be asked to complete a baseline demographics and health behaviors survey, which includes the Pittsburgh Sleep Quality Index, a dietary habits and physical activity questionnaire, and quality of life and personal health questionnaire (S3) [27]. At the follow-up visit, participants will complete a shorter version of the same survey (S3). From this point forward, the baseline and follow-up visits will progress in the same order. A trained researcher will record participant body temperature, pulse rate, respiration rate, and resting blood pressure. Blood pressure will be measured using a calibrated instrument after 5 minutes of quiet sitting using an appropriately sized arm cuff. The researcher will then collect body weight and measures of body composition (fat-free mass, fat mass, bone mass) using a validated bioelectrical impedance analysis (BIA) scale (Tanita TBF-410GS Body Composition Analyzer). Upon returning to the clinic room, venous blood samples, specifically, a complete metabolic panel, complete blood count with differential, fasting lipid panel, and hemoglobin A1c will be obtained (Table 1). Additional blood will be drawn for biomarker analysis (Table 2). A total of 30 mL of fasted blood will be drawn each visit by a certified phlebotomist in the OCTRI outpatient clinic. A trained medical professional will then administer the Unified Huntington's Disease Rating Scale (UHDRS). The UHDRS is a clinical rating scale developed to assess four domains of HD – motor function, cognitive function, behavioral abnormalities, and functional capacity. Specifically, the assessment consists of a 15-item motor exam, the Symbol Digit Modalities Test, the Stroop Interference Test, a verbal fluency test, an interview-style assessment developed to rate frequency and severity of behavioral symptoms, and an evaluation of personal function which grades independence and capacity to complete daily tasks [28].

At the baseline visit, once the UHDRS is completed, participants will be instructed how to download and use the SnapCalorie™ phone application. They will also be provided a scale for at-home body weight measurements and be informed of the TRE intervention. They will be instructed to maintain a 6-8-hour eating window every day for 12 weeks after completion of the lead-in period. Participants will be allowed to self-select the timing of the eating window; once selected, they should attempt to maintain the schedule every day for the duration of the study. Outside of that window, for the remaining 16-18 hours of day/night, participants will

| | STUDY PERIOD | | | | | |
|---|---|---|---|---|---|---|
| | Enrollment | Allocation | Post-allocation | | | Close-out |
| **TIMEPOINT** | *-t₁* | **0** | *t₀* | *t₁* (week 1) | *t₂* (week 13) | *t₂* (week 13) |
| **ENROLLMENT:** | | | | | | |
| *Eligibility screen* | X | | | | | |
| *Informed consent* | X | | | | | |
| *Allocation* | | X | | | | |
| **INTERVENTIONS:** | | | | | | |
| *Time-restricted eating* | | | | ●———————● | | |
| **ASSESSMENTS:** | | | | | | |
| *Telephone Screening evaluation* | X | | | | | |
| *MoCA* | | | X | | | X |
| *Pregnancy Test* | | | X | | | |
| *In-Lab Questionnaire* | | | X | | | X |
| *BIA* | | | X | | | X |
| *Vital Signs* | | | X | | | X |
| *Venipuncture Blood Draw* | | | X | | | X |
| *UHDRS* | | | X | | | X |
| *Daily Meal Timing and Sleep Survey* | | | | ●———————● | | |
| *SnapCalorie™ Meal Tracking* | | | | ●———● | | |
| *Weekly Survey* | | | | ●———● | | |
| *Telephone Check-In* | | | | ●———● | | |

**Fig 3. Schedule of enrollment, interventions, and assessments.** MoCA: Montreal Cognitive Assessment; BIA: bioelectrical impedance analysis; UHDRS: Unified Huntington's Disease Rating Scale.

be asked not to consume calorie-containing food or drink. Beverages without calories, such as water, black coffee or tea, will be allowed; however, zero-calorie beverages with artificial sweeteners, such as diet soda, will not. Participants will be instructed to consume plenty of water and electrolytes. Participants will also be provided a document with information about general healthy diet and exercise habits, in addition to instructions about the TRE protocol and a brief overview of each study visit before they depart from the clinic (S3).

During the 1-week lead-in period after the baseline visit, participants will be asked to use the SnapCalorie™ phone application and to complete the daily caloric timing and sleep survey to log at least 7 days of their habitual dietary and sleep habits (S3). Daily calories and macronutrients will be calculated and used to provide dietary caloric goals during the 12-week TRE intervention.

During the lead-in period and the TRE phase, a daily electronic survey will be sent to each participant via text message or email, depending on participant preference, in the evening at 19:00h. Participants will be asked to report the time of day they started and stopped eating, and their sleep duration and quality from the previous night (S3). Twice per week in the TRE phase, participants will be asked to use the SnapCalorie™ phone application to capture the calories consumed in a given day. A text or email reminder will be sent in the morning at 08:00h on the days that the participants are asked to record their calories. Using the application, participants will be asked to take photos of each caloric event and report estimated serving sizes. The application uses artificial intelligence technology and manually entered nutrition information to analyze caloric composition which will be specifically used to estimate caloric content, and daily consumed fats, carbohydrates, proteins, saturated fats, cholesterol, sodium, fiber, and sugar [29]. Photos of meals will be used to describe the quality and types of foods being consumed. The application also time-stamps images and will be used as an additional method to track the eating window during the TRE intervention. Participants will also be asked to complete a weekly survey to measure self-reported body weight, mood, and physical activity (S3). The survey will be completed via Qualtrics™. Researchers will send a reminder text or email each week with a link to the survey. Every other week during

**Table 1. Safety blood parameters.**

| Metabolic Panel | Hematology Panel | Lipid Panel |
|---|---|---|
| Sodium | White cell count | Total Cholesterol |
| Potassium | Red cell count | Triglycerides |
| Chloride | Hemoglobin | High Density Lipoprotein |
| Total CO2 | Hematocrit | Low Density Lipoprotein |
| Anion Gap | Mean Corpuscular Volume | Very Low Density Lipoprotein |
| Blood Urea Nitrogen (BUN) | Mean Corpuscular Hemoglobin | Non-High Density Lipoprotein |
| Creatinine | Mean Corpuscular Hemoglobin Concentration | |
| Estimated Glomerular Filtration Rate (EGFR) | Red Cell Distribution Width | |
| Glucose | Platelet Count | |
| Calcium | Mean Platelet Volume | |
| Aspartate Aminotransferase (AST) | Nucleated Red Blood Cell % | |
| Alanine Aminotransferase (ALT) | Neutrophil % and # | |
| Alkaline phosphatase | Lymphocyte % and # | |
| Total Bilirubin | Monocyte % and # | |
| Total Protein | Eosinophil % and # | |
| Albumin | Basophil % and # | |
| BUN/Creatinine Ratio | Immunoglobulin % and # | |
| Globulin Level | | |

**Table 2. Protocol measures description.**

| Outcome measures | | |
| --- | --- | --- |
| Measure | Domain (outcome) | Description |
| Protocol adherence | Feasibility (primary) | The percent of days participants can successfully restrict the eating window to 6-8 hours, as tracked through self-report and time-stamped meal logs, will be calculated for each participant during the 12 weeks of TRE [24]. |
| Fat-free body mass | Safety (secondary) | Using bioelectrical impedance analysis, body composition, and specifically, fat-free body mass, will be measured before and after the TRE intervention at the baseline and follow-up visits [31]. |
| Daily eating period | Feasibility (secondary) | The timeframe that participants consume calories within (from first consumption to last in the 24-hour day) will be measured through retrospective survey analysis and during the lead-in week prior to the TRE intervention via self-report and time-stamped meal logs. This will be compared to the average duration of the eating period during the 12 weeks of TRE, where participants are asked to limit the period to only 6-8 hours, while fasting the remainder of the 24-hour day [24]. |
| Plasma neurofilament light protein (NfL) | Biomarker effects (secondary) | NfL is a protein marker of neurodegeneration that can be detected peripherally in the blood. Levels of NfL will be measured via ELISA from blood drawn before and after the TRE intervention at the baseline and follow-up visits. [26,32]. |
| Plasma glial fibrillary acidic protein (GFAP) | Biomarker effects (secondary) | GFAP is a protein marker of neurodegeneration that can be detected peripherally in the blood. Levels of GFAP will be measured via ELISA from blood drawn before and after the TRE intervention at the baseline and follow-up visits [33]. |
| Composite Unified Hunting-ton's Disease Rating Scale (cUHDRS) | Clinical effects (secondary) | The UHDRS is a clinical rating scale developed to assess four domains of HD – motor function, cognitive function, behavioral abnormalities, and functional capacity [28]. The UHDRS will be administered before and after the TRE intervention at the baseline and follow-up visits. |
| Safety blood panels | Safety (other) | A comprehensive metabolic panel (CMP), complete blood count (CBC) with differential, lipid panel, and hemoglobin A1c will be drawn at the baseline and follow-up visits. |
| Body weight | Safety (other) | At the baseline visit, participants will be given a scale to measure at-home body weight. They will be asked to self-report their body weight in the weekly survey. Every other week during the study, participants will be contacted by telephone to discuss any at-home body weight changes. |
| Dietary composition | Exploratory (other) | At the baseline visit and follow-up visit, participants will be asked to complete a dietary composition and habits questionnaire (S3). Twice per week, and each day during the lead-in period, participants will be asked to use the SnapCalorie™ phone application to capture the calories consumed in a given day. At the baseline visit, participants will be instructed how to download and use the phone application. A text or email reminder will be sent in the morning at 08:00 h on the days that the participants are asked to record their caloric intake. Using the application, participants will be asked to take photos of each meal, manually log foods, and report estimated serving sizes. Results will be used to estimate caloric intake, and daily consumed fats, carbohydrates, and proteins [29]. The application also time-stamps images and will be used as an additional method to track the eating window during the TRE intervention. |
| Sleep | Exploratory (other) | At the baseline and follow-up visit, participants will be asked to complete the Pittsburgh Sleep Quality Index questionnaire (S3) [27]. In a daily Qualtrics™ survey, participants will be asked to self-report sleep onset and offset from which we can calculate daily duration. In a weekly Qualtrics™ survey, participants will be asked to complete the Epworth Sleepiness Scale questionnaire [34]. |
| Mood | Exploratory (other) | In a weekly Qualtrics™ survey, participants will be asked to self-report various aspects of their mood. See the "Weekly Survey" document in the supplemental information for additional details. |
| Cognition | Exploratory (other) | The Montreal Cognitive Assessment (MoCA) is a validated test used to measure cognitive function [35]. It will be administered before and after the TRE intervention at the baseline and follow-up visits. |
| Physical activity | Exploratory (other) | In a weekly Qualtrics™ survey, participants will be asked to self-report the amount and type of exercise they engaged in during the previous seven days. See the "Weekly Survey" document in the supplemental information for additional details. |
| Mitochondrial function | Exploratory (other) | Using cryopreserved peripheral blood mononuclear cells (PBMCs) isolated from participant whole blood samples drawn at the baseline and follow-up visits, mitochondrial function and electron transport chain activity will be assessed with the SeahorseXF analyzer as an exploratory mechanism-of-action measure of TRE [36,37]. |

the 12-week TRE phase of the study, participants will be contacted by telephone to discuss at-home body weight changes and any adherence issues or adverse events.

## Outcome measures

**Primary outcome measure.** The primary outcome measure will be the percent of days participants adhere to the TRE diet. Adherence, measured as the number of days participants can successfully limit the eating window to 6-8 hours as tracked through self-reported surveys and time-stamped meal logs, will be calculated for each participant during the 12 weeks of TRE. If ≥80% of participants can eat within the pre-specified window ≥ 80% of the trial days, then the intervention will be considered feasible.

**Secondary outcome measures.** The secondary outcomes will encompass various domains, including assessments of safety, feasibility, and efficacy. The change from baseline in fat-free body mass, measured using bioelectrical impedance analysis at the baseline and follow-up visits, will be used to evaluate the safety of the intervention. The change from baseline in the daily eating period will serve as an additional measure of feasibility. The timeframe that participants consume calories within (from first consumption to last in the 24-hour day) will be measured through retrospective survey analysis and during the lead-in week prior to the TRE intervention via self-report and time-stamped meal logs. This will be compared to the average duration of the eating period during the 12 weeks of TRE, where participants are asked to limit the period to only 6-8 hours, while fasting the remainder of the 24-hour day. The change from baseline in plasma NfL and glial fibrillary acidic protein (GFAP) – markers of neurodegeneration – will be used to evaluate the physiologic effects of the intervention. We will also compare plasma NfL concentrations to natural history control data to contextualize the observed results [26,30]. Lastly, the change from baseline in the Composite Unified Huntington's Disease Rating Scale (cUHDRS) score will be used to explore clinical effects of the intervention.

The cUHDRS includes four parts. The first assesses motor function using 31 items with a 5-point ordinal scale ranging from 0-4 with the highest score indicating inability to perform the motor task. Part two examines cognitive function using three items – Verbal Fluency Test, Symbol Digit Modalities Test, and Stroop Interference Test – with higher scores indicating better cognitive performance. Part three evaluates behavior using 10 items with a 5-point ordinal scale ranging from 0-4 with the highest score indicating severe behavioral symptoms, and 4 items requiring the evaluator to answer yes/no questions about the overall clinical impression with respect to the participant showing clinical evidence of confusion, dementia, depression and requiring pharma-cotherapy. Part four assesses functional capacity and is divided into three sections: Huntington's Disease Functional Capacity Scale (HDFCS) is reported as the Total Functional Capacity Score (TFC) which has a total of 25 yes/no questions assessing the total functional capacity of the indi-vidual; Independence Scale rated from 10 to 100 with higher scores indicating better functioning than lower scores; and Functional Capacity using 5 items with a 4-point ordinal scale ranging from 0 to 3 with the highest score indicating higher functional capacity.

**Other data to be collected.** Additional measures will be obtained to further characterize the safety and explore the effects of TRE in this population. See Table 2 for a full description of each measure.

## Data management

While the study is active, all PHI and study data will be stored in a database housed on a secure OHSU server. PHI will be accessible only to investigators, and research staff listed on the protocol. No PHI will be stored or transferred via USB or other portable drives. We will obtain a waiver of the HIPPA authorization requirement for PHI collection at the time of the phone and survey screen.

At the initial phone-screen, all participants will be assigned a unique code that will be used to identify them on documents residing outside of the secure database. Upon enrollment, participant study data will be recorded by the research coordinator and staff. The data will be verified by the staff members, by double data entry and visual verification. Any discrepancies will be evaluated and manually reconciled by the verifier. Study data will be stored in password protected and OHSU encrypted OneDrive and Qualtrics™ databases and will be identified only by the unique code assigned at the initial phone screen. Only persons listed on the IRB approved protocol will be given access to these databases.

All paper files (e.g., signed consent forms) will be stored in locked filing cabinets in restricted access offices at OHSU. Original records will be retained as the source document. Access to these files will be limited to study personnel listed on the IRB approved protocol.

Blood samples will be identified only by the code assigned at the initial phone screen. Processing and handling of blood samples will be done by the OCTRI nursing and core lab staff. Samples will be stored and maintained at -80 degrees C. Access to study samples at OHSU will be limited to OCTRI core lab staff and study personnel listed on the IRB approved protocol. Coded biological samples may be sent to the Quanterix Corporation Laboratory for analysis.

## Statistical analysis

The primary goals of this study are to assess if TRE in persons with premanifest and early manifest HD is safe, feasible, and potentially efficacious. Analyses will focus on evaluating effect size and variability in our measures to help us identify important safety and feasibility trends. Our primary outcome will be the percentage of days participants are adherent to the TRE intervention. The percent of adherent days will be calculated for each participant during the 12 weeks of TRE. If ≥ 80% of participants can eat within the pre-specified window ≥ 80% of the trial days, then the intervention will be considered feasible.

Additionally, we will measure several other secondary outcomes to assess safety, feasibility, biomarker dynamics, and clinical effects. The main safety outcome will be the percent change in body weight and fat-free mass. Differences in measures of vital signs, and safety labs will also be examined between the baseline and follow-up visits as additional safety measures. The average daily eating period at baseline, calculated using retrospective survey and lead-in period data, will be compared to the average recorded daily eating period during the 12 weeks of TRE. Finally, differences in measures of blood biomarkers and UHDRS clinical scores will be examined between the baseline and follow-up visits. To evaluate outcomes that are only measured at baseline and at the 12-week follow-up, we will utilize a combination of paired samples t-tests to compare pre- versus post-intervention values, and descriptive statistics to assess measures of central tendency (means, medians, etc.) along with measures of variability (standard deviation, 95% confidence intervals). For outcomes that include more than two timepoints, in addition to descriptive statistics, we will use graphical representations of the longitudinal data (e.g., line plots) to help us assess trends and between participant variation. The pre- versus post-interventional analyses and variance estimates will be critical in helping determine appropriate power and sample size calculations for a future larger-scale clinical trial.

## Safety considerations

To the best of our knowledge, TRE has not been evaluated clinically in persons with HD. Therefore, we must rely on previous studies that have explored the intervention in other populations to inform our safety considerations. Applying a dietary intervention that has been associated with weight loss is concerning in a disease like HD, which is known for

causing decreases in body weight [23]. It is reassuring, however, that weight loss associated with TRE is primarily found in studies with overweight and obese subjects and is thought to be strictly related to a decrease in caloric intake rather than the exercise of fasting itself [38,39]. Additionally, a randomized crossover trial that specifically sought to assess the safety and tolerability of TRE (16-h fast/8-h feed) in healthy midlife and older adults for six weeks found the dietary intervention had no influence on body mass, lean mass, bode density, or nutrient intake [24]. It is noteworthy that subjects in this trial were advised how to maintain healthy daily caloric intake within the restricted eating window. This information underlies our reasoning for a baseline retrospective dietary survey and lead-in period which utilizes a phone application to capture and record dietary composition and daily caloric intake in our participants. Using these data, we can more accurately provide and recommend macronutrient and caloric targets for our participants to aim for during their restricted eating window in an effort to minimize weight change. Participants will also be provided a scale to take at-home body weight measures and will be asked to report weight in a weekly survey. Study personnel will monitor these results and discuss findings at the alternate week phone-call check-ins. These conversations will also provide participants the opportunity to discuss adverse effects, difficulties with adherence, or questions regarding the intervention. If a participant loses more than 10% of their total body weight, they will be asked to stop the TRE diet and return to their prior dietary habits. Participants will be allowed to receive concomitant health care for HD or other medical conditions during the trial; however, we will ask that changes from their baseline health survey – including active medical therapies, recent surgeries, and newly diagnosed medical conditions – be reported to the study team during the phone-call check-ins.

Additionally, a data safety monitory plan has been approved by the OHSU IRB. The plan states that the research coordinator and staff will be recording study data. The data will be verified by the staff members, by double data entry and visual verification. Any discrepancies will be evaluated and manually reconciled by the verifier. The principal investigator will verify that procedures are being conducted according to the protocol by conducting a monthly meeting which will review all procedures done. Any unanticipated problems will be identified by study staff in real time, reviewed by investigators and reported to the IRB as appropriate according to institutional policy and applicable regulations. Study personnel will report adverse events and unanticipated problems to the OHSU IRB via the electronic IRB (eIRB) system, within the timeframes stipulated by OHSU.

## Ethics and dissemination

The study design and the protocol were submitted and approved by the OHSU Institutional Review Board (approval study #00026970) on August 13, 2024. When implementing the study, we will follow the guidelines of the World Health Organization Code of Conduct for Responsible Research and the World Medical Association Code of Ethics Declaration of Helsinki to ensure all results are reliable and unbiased, and that the rights, welfare, integrity, and confidentiality of all participants are protected at all times. Public access to the study protocol has been ensured via the publication and registration of the trial protocol at ClinicalTrials.gov (NCT06490367). Any protocol modifications (e.g., changes to eligibility criteria, outcomes, analyses etc.) will be reported and tracked by the OHSU IRB and the public registry for the study (ClinicalTrials.gov). The study results will be posted to ClinicalTrials.gov and are expected to be published in a peer-reviewed scientific journal. The final trial dataset will be deidentified and made available for public access on Mendeley Data, a free and secure cloud-based repository.

### Anticipated timeline

The recruitment of participants began in September 2024 and will continue until February 2024 or earlier if enrollment targets are achieved by a prior date. Baseline data will be collected at the onset of a participant's enrollment. Follow-up will occur roughly three months after the baseline visit and will take place within one week after the TRE intervention is completed. The trial will finish no later than June 2025.

## Discussion

This pilot study represents the crucial first step in the clinical exploration and evaluation of TRE's therapeutic potential as a lifestyle intervention for HD. Considering that TRE can dramatically alter the course of the disease in multiple HD animal models, we are confident it is time to begin a robust clinical assessment of the intervention in humans [16]. To lay the foundation for large scale trials powered to evaluate clinical efficacy, we must first understand if it is feasible, safe, and has any physiologic effect on markers of disease progression.

We expect the TRE intervention applied in this trial will be feasible, tolerable, and safe in persons with early-stage HD [24]. We also expect to see trends toward improvement in biomarkers of neurodegeneration and mitochondrial function [16]. Given the small sample size and short-term nature of the study, we do not expect to observe statistically significant changes in pre-intervention versus post-intervention clinical UHDRS measures.

Another point of discussion regards the decision to allow participants to self-select the timing of the eating window. The exact benefits associated with the timing of TRE induction remain to be fully understood. It is true that early-TRE (eating breakfast and lunch) has been shown to potentially enhance specific circadian and metabolic benefits; however, delayed-TRE (eating lunch and dinner) may better coincide with natural patterns of hunger and social routines, potentially promoting mental and emotional health [40]. It is also worth emphasizing that the majority of the mechanisms underlying the therapeutic potential of TRE in the context of HD, including increased autophagic activity, BDNF production, and mitochondrial biogenesis, appear to depend more on the length of the fasting perioid, rather than its timing within the 24-hour day [16]. Considering these observations and our primary goal to assess the feasibility of TRE in this population, we think it is reasonable to allow participants to self-select the timing of the 8-hour eating window.

Despite the many strengths of this study design, a few limitations exist. To minimize participant time-commitment and study burden we plan to rely heavily on survey-based self-reported data, specifically regarding adherence and tracking of the eating window. Participant recollection, accuracy in reported information, and honesty will determine these outcomes of the study. The image-based collection methods that will be used twice a week during the intervention will provide time-stamped data that will help validate survey data. Other limitations that are inherent to an open-label single-arm trial design such as a lack of a comparison group, placebo effect, and presence of confounders will exist; although, since the goals of the pilot study are to evaluate the feasibility and safety of TRE rather than to prove its clinical efficacy, we are confident in this design choice given our available resources. We expect it will provide sufficient evidence to support the primary and secondary outcome measures. The data generated in this project will guide alternative approaches to be used in future studies, i.e., increasing the number of participants, adding a control group, and lengthening the intervention timeline.

To the best of our knowledge, this project will be the first interventional dietary study in persons with HD and will be one of the earliest interventional clinical trials to assess the

effects of TRE in a neurodegenerative disease. Our focus on bioenergetics and biomarkers of neurodegeneration makes this study particularly distinct from an existing protocol examining the effects of fasting on inflammation and the gut microbiome in Parkinson's disease [41]. Here, we will integrate validated basic science, nutritional status, and clinical outcomes to evaluate the safety, feasibility, and preliminary efficacy of TRE in HD. It will also incorporate novel methods for dietary composition tracking and biomarker analyses capable of elucidating components of TRE's mechanism of action. These methods could prove useful for a broad array of future trials in nutritional or metabolic interventional studies of various disease states. This project will specifically generate pilot data that will lay the foundation for future large-scale clinical trials powered to evaluate the long-term efficacy of TRE in HD.

## Supporting information

**File S1. SPIRIT 2013 checklist – recommended items to address in a clinical trial protocol.**
(DOCX)

**File S2. Oregon health and science university institutional review board approved protocol.**
(DOCX)

**File S3. In-lab Baseline Survey; In-lab Follow-up Survey; Daily QualtricsTM Survey; Weekly QualtricsTM Survey.**
(DOCX)

## Acknowledgements

The authors are grateful to Dr. Cynthia Morris and the Oregon Clinical and Translational Research Institute for input on the trial design. We additionally acknowledge Dr. Barbara Brumbach and Dr. Alicia Johnson for assistance with the development of the statistical analysis plan. Dr. Hiller and Dr. Neilson are also part of the Pacific Northwest Parkinson's Disease Research, Education, and Clinical Center (PADRECC) within the Department of Veterans Affairs.

## Author contributions

**Conceptualization:** Russell G. Wells.

**Funding acquisition:** Russell G. Wells, Amie L. Hiller.

**Investigation:** Russell G. Wells, Amie L. Hiller.

**Methodology:** Russell G. Wells, Lee E. Neilson, Andrew W. McHill, Amie L. Hiller.

**Project administration:** Russell G. Wells.

**Resources:** Russell G. Wells.

**Software:** Russell G. Wells.

**Supervision:** Russell G. Wells, Amie L. Hiller.

**Validation:** Russell G. Wells.

**Visualization:** Russell G. Wells.

**Writing – original draft:** Russell G. Wells.

**Writing – review & editing:** Russell G. Wells, Lee E. Neilson, Andrew W. McHill, Amie L. Hiller.

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
