## [Decision Letter · Decision Letter 0]

13 Nov 2024

PONE-D-24-40480Time-restricted eating in early-stage Huntington’s disease: A 12-week interventional clinical trial protocolPLOS ONE

Dear Dr. Wells,

Thank you for submitting your manuscript to PLOS ONE. After careful consideration, we feel that it has merit but does not fully meet PLOS ONE’s publication criteria as it currently stands. Therefore, we invite you to submit a revised version of the manuscript that addresses the points raised during the review process.

We look forward to receiving your revised manuscript.

Kind regards,

Prof. Jussi Sipilä

M.D., Ph.D., B.Soc.Sci.

Academic Editor

PLOS ONE

Journal Requirements:

2. Thank you for stating the following financial disclosure: [This work is supported by a PCO pilot award (to R.G.W), VA CSR&D CDA2 IK2 CX00253-01A1 (to L.E.N.), the Parkinson’s Disease Research, Education, Clinical Center at the VA Portland Health Care System, and by the National Center for Advancing Translational Sciences, National Institutes of Health, through Grant Award Number TL1TR002371 (to R.G.W). The content is solely the responsibility of the authors and does not necessarily represent the official views of the NIH.]

4. Thank you for stating the following in the Acknowledgments Section of your manuscript: [The authors are grateful to Dr. Cynthia Morris and the Oregon Clinical and Translational Research Institute, which is supported by the National Center for Advancing Translational Sciences, National Institutes of Health, through Grant Award Number UL1TR002369, for input on the trial design. We additionally acknowledge Dr. Barbara Brumbach and Dr. Alicia Johnson for assistance with the development of the statistical analysis plan. Dr. Hiller and Dr. Neilson are also part of the Pacific Northwest Parkinson’s Disease Research, Education, and Clinical Center (PADRECC) within the Department of Veterans Affairs.]

Please remove any funding-related text from the manuscript and let us know how you would like to update your Funding Statement. Currently, your Funding Statement reads as follows: [This work is supported by a PCO pilot award (to R.G.W), VA CSR&D CDA2 IK2 CX00253-01A1 (to L.E.N.), the Parkinson’s Disease Research, Education, Clinical Center at the VA Portland Health Care System, and by the National Center for Advancing Translational Sciences, National Institutes of Health, through Grant Award Number TL1TR002371 (to R.G.W). The content is solely the responsibility of the authors and does not necessarily represent the official views of the NIH.]

5. We noted in your submission details that a portion of your manuscript may have been presented or published elsewhere. [Yes, figure one has previously been published by our group in a recent journal article. This figure outlines key components described in the introduction section of the current manuscript under consideration.]

Please clarify whether this publication was peer-reviewed and formally published. If this work was previously peer-reviewed and published, in the cover letter please provide the reason that this work does not constitute dual publication and should be included in the current manuscript.

6. Your abstract cannot contain citations. Please only include citations in the body text of the manuscript, and ensure that they remain in ascending numerical order on first mention.

7. Your ethics statement should only appear in the Methods section of your manuscript. If your ethics statement is written in any section besides the Methods, please move it to the Methods section and delete it from any other section. Please ensure that your ethics statement is included in your manuscript, as the ethics statement entered into the online submission form will not be published alongside your manuscript.

8. When completing the data availability statement of the submission form, you indicated that you will make your data available on acceptance. We strongly recommend all authors decide on a data sharing plan before acceptance, as the process can be lengthy and hold up publication timelines. Please note that, though access restrictions are acceptable now, your entire data will need to be made freely accessible if your manuscript is accepted for publication. This policy applies to all data except where public deposition would breach compliance with the protocol approved by your research ethics board. If you are unable to adhere to our open data policy, please kindly revise your statement to explain your reasoning and we will seek the editor's input on an exemption. Please be assured that, once you have provided your new statement, the assessment of your exemption will not hold up the peer review process.

Additional Editor Comments:

Looking good. Please also see and respond to the comments below:

This study protocol intends on examining the feasibility and safety of time-restricted eating (TRE) in 25 patients with Huntington’s disease.

Conclusion

Overall, this study protocol is well thought through and although the number of participants is small, this study will bring exciting results on the impact of nutrition and lifestyle interventions in neurodegenerative diseases.

Comments

· Figure 1 and table 1 are overlapping, it will be more beneficial to integrate the information of the table into the figure and have a better overview

· The self-selection of the eating window light is not ideal to see beneficial changes in a small study sample, however, I agree that it might not be feasible to impose an eating window

· Exclusion criteria: maybe add history of eating disorder as exclusion criteria

· Additional sample collection during the 12 weeks would be of great interest to track possible changes induced by TRE, rather than only checking before and after

192 Are not weight stable…

207 NfL abbreviation only introduced in line 308

243 Only these values? Maybe include an extensive list of what will be measured

247 What other questionnaires are included? Well-being etc. Include list with questionnaires and references

265 Were there clear instructions on zero-calorie beverages with sweeteners?

281 Why only kcal and macronutrient composition? The quality of the food (e.g. highly processed) and the micronutrient composition are of great interest and should be considered as well.

455 This is not the first intervention trial looking ate the effects of TRE in neurodegenerative diseases: Hansen B, Laczny CC, Aho VTE, et al Protocol for a multicentre cross-sectional, longitudinal ambulatory clinical trial in rheumatoid arthritis and Parkinson’s disease patients analysing the relation between the gut microbiome, fasting and immune status in Germany (ExpoBiome) BMJ Open 2023;13:e071380. doi: 10.1136/bmjopen-2022-071380

TABLE 1 Why are no dietary questionnaires and sleep etc. tracked on baseline visit?

Reviewers' comments:

Reviewer's Responses to Questions

**Comments to the Author**

1. Does the manuscript provide a valid rationale for the proposed study, with clearly identified and justified research questions?

Reviewer #1: Yes

2. Is the protocol technically sound and planned in a manner that will lead to a meaningful outcome and allow testing the stated hypotheses?

Reviewer #1: Yes

3. Is the methodology feasible and described in sufficient detail to allow the work to be replicable?

Reviewer #1: Yes

4. Have the authors described where all data underlying the findings will be made available when the study is complete?

Reviewer #1: Yes

5. Is the manuscript presented in an intelligible fashion and written in standard English?

Reviewer #1: Yes

6. Review Comments to the Author

You may also provide optional suggestions and comments to authors that they might find helpful in planning their study.

Reviewer #1: This is a well designed and well written protocol for time-restricted eating in early-stage Huntington’s disease. However, I have a few comments.

1. It is a single arm study. Therefore, it is useful to cite any matching reference in the literature for placebo to compare the results to get an initial preliminary idea to proceed with large two arm trials.

2. Participants will be allowed to self-select the timing of the eating window. I understand that it is to study the feasibility of the intervention. However, is not it better to recommend participants for internment fasting window from late afternoon until next morning to match with the circadian rhythm of the body?

3. In line 116: separate the objectives as primary, secondary, exploratory etc.

4. Minimum age for inclusion is at least 21 years. Why not 18-years?

5. How likely to have any participants who are not fluent in English and needs translator. If so, are they going to be excluded?

7. PLOS authors have the option to publish the peer review history of their article (what does this mean? ). If published, this will include your full peer review and any attached files.

**Do you want your identity to be public for this peer review?** For information about this choice, including consent withdrawal, please see our Privacy Policy .

Reviewer #1: **Yes: ** Dr Shah-Jalal Sarker

---

## [Author Response · Author response to Decision Letter 1]

30 Dec 2024

Response to Reviewers PONE-D-24-40480

We appreciate the feedback and opportunity to respond to these comments to strengthen our manuscript. Unedited comments are left in bold and our responses are below. If necessary, the highlight function has been used to identify textual changes without losing context of surrounding material.

RESPONSE: We have amended the text to meet PLOS ONE’s style requirements, specifically making font adjustments to the title, corresponding author contact information, and level 1, 2, and 3 headings.

2. Thank you for stating the following financial disclosure: [This work is supported by a PCO pilot award (to R.G.W), VA CSR&D CDA2 IK2 CX00253-01A1 (to L.E.N.), the Parkinson’s Disease Research, Education, Clinical Center at the VA Portland Health Care System, and by the National Center for Advancing Translational Sciences, National Institutes of Health, through Grant Award Number TL1TR002371 (to R.G.W). The content is solely the responsibility of the authors and does not necessarily represent the official views of the NIH.]

RESPONSE: The funders are institutions and institutional grants. Some of the authors are employees of the institutions but there were no institutional representatives beyond the authors of the paper that were involved in the decision to publish or preparation of the manuscript. The acknowledgement section lists other institutional employees who were involved in study design.

4. Thank you for stating the following in the Acknowledgments Section of your manuscript: [The authors are grateful to Dr. Cynthia Morris and the Oregon Clinical and Translational Research Institute, which is supported by the National Center for Advancing Translational Sciences, National Institutes of Health, through Grant Award Number UL1TR002369, for input on the trial design. We additionally acknowledge Dr. Barbara Brumbach and Dr. Alicia Johnson for assistance with the development of the statistical analysis plan. Dr. Hiller and Dr. Neilson are also part of the Pacific Northwest Parkinson’s Disease Research, Education, and Clinical Center (PADRECC) within the Department of Veterans Affairs.]

Please remove any funding-related text from the manuscript and let us know how you would like to update your Funding Statement. Currently, your Funding Statement reads as follows: [This work is supported by a PCO pilot award (to R.G.W), VA CSR&D CDA2 IK2 CX00253-01A1 (to L.E.N.), the Parkinson’s Disease Research, Education, Clinical Center at the VA Portland Health Care System, and by the National Center for Advancing Translational Sciences, National Institutes of Health, through Grant Award Number TL1TR002371 (to R.G.W). The content is solely the responsibility of the authors and does not necessarily represent the official views of the NIH.]

RESPONSE: The text in the Acknowledgements Section has been amended to now read, “The authors are grateful to Dr. Cynthia Morris and the Oregon Clinical and Translational Research Institute for input on the trial design. We additionally acknowledge Dr. Barbara Brumbach and Dr. Alicia Johnson for assistance with the development of the statistical analysis plan. Dr. Hiller and Dr. Neilson are also part of the Pacific Northwest Parkinson’s Disease Research, Education, and Clinical Center (PADRECC) within the Department of Veterans Affairs.” We moved the funding-related text to the Funding Statement and have included this amended statement in the cover letter.

5. We noted in your submission details that a portion of your manuscript may have been presented or published elsewhere. [Yes, figure one has previously been published by our group in a recent journal article. This figure outlines key components described in the introduction section of the current manuscript under consideration.]

Please clarify whether this publication was peer-reviewed and formally published. If this work was previously peer-reviewed and published, in the cover letter please provide the reason that this work does not constitute dual publication and should be included in the current manuscript.

RESPONSE: Figure 2 was previously created and published by our group in a peer-reviewed journal article in Translational Neurodegeneration under an open-access license with BMC Springer Nature where the reproduction of figures is permitted with proper citation of the original source. We include clear attribution and citation of the original source of this previously reported material. Lastly, we believe the figure enhances the introduction section of this manuscript by providing a clear summary of the mechanisms underlying the therapeutic potential of the intervention we are investigating.

Citation: Wells RG, Neilson LE, McHill AW, Hiller AL. Dietary fasting and time-restricted eating in Huntington's disease: therapeutic potential and underlying mechanisms. Transl Neurodegener. 2024;13(1):17 doi: 10.1186/s40035-024-00406-z.

Link: https://translationalneurodegeneration.biomedcentral.com/articles/10.1186/s40035-024-00406-z

6. Your abstract cannot contain citations. Please only include citations in the body text of the manuscript, and ensure that they remain in ascending numerical order on first mention.

RESPONSE: The abstract has been reviewed, and we ensure there are no citations present. All citations are included in the body text of the manuscript and are in ascending numerical order by first mention.

7. Your ethics statement should only appear in the Methods section of your manuscript. If your ethics statement is written in any section besides the Methods, please move it to the Methods section and delete it from any other section. Please ensure that your ethics statement is included in your manuscript, as the ethics statement entered into the online submission form will not be published alongside your manuscript.

RESPONSE: We ensure the ethics statement only appears in the Methods section of the manuscript. It can be found under the level 2 subheading of the Methods section, labelled, “Ethics and dissemination.”

8. When completing the data availability statement of the submission form, you indicated that you will make your data available on acceptance. We strongly recommend all authors decide on a data sharing plan before acceptance, as the process can be lengthy and hold up publication timelines. Please note that, though access restrictions are acceptable now, your entire data will need to be made freely accessible if your manuscript is accepted for publication. This policy applies to all data except where public deposition would breach compliance with the protocol approved by your research ethics board. If you are unable to adhere to our open data policy, please kindly revise your statement to explain your reasoning and we will seek the editor's input on an exemption. Please be assured that, once you have provided your new statement, the assessment of your exemption will not hold up the peer review process.

RESPONSE: Our apologies, we made an error when completing the data availability statement within the submission form. We have amended the statement to read, “No data were derived for this publication.”

RESPONSE: The reference list has been reviewed, and we confirm that each citation is complete, correct, and relevant. To the best of our knowledge, none of the cited papers have been retracted.

Additional Editor Comments:

Looking good. Please also see and respond to the comments below:

This study protocol intends on examining the feasibility and safety of time-restricted eating (TRE) in 25 patients with Huntington’s disease.

Conclusion

Overall, this study protocol is well thought through and although the number of participants is small, this study will bring exciting results on the impact of nutrition and lifestyle interventions in neurodegenerative diseases.

RESPONSE: Thank you very much for the positive feedback on the design of this study and its potential impact on the field. We believe the comments made below will make this a stronger protocol manuscript.

Comments

· Figure 1 and table 1 are overlapping, it will be more beneficial to integrate the information of the table into the figure and have a better overview

RESPONSE: We have integrated the information that was in Table 1 into Figure 1.

· The self-selection of the eating window light is not ideal to see beneficial changes in a small study sample, however, I agree that it might not be feasible to impose an eating window

RESPONSE: This is an understandable consideration. We agree, however, that a self-selected eating window will maximize feasibility in this population. We have added more information on our reasoning for this decision in the “Discussion” section. We may consider controlling for a specific eating window in future studies.

· Exclusion criteria: maybe add history of eating disorder as exclusion criteria

RESPONSE: This is an important consideration, and we believe this falls within the current exclusion point 11; however, we have amended the text to make this point more explicit. It now reads, “Subjects ineligible to participate in this study are persons who: 11. Have a clinically significant medical (including a current or prior eating disorder), surgical, laboratory, or behavioral abnormality which in the judgement of the site Investigator makes the subject unsuitable for the study.”

· Additional sample collection during the 12 weeks would be of great interest to track possible changes induced by TRE, rather than only checking before and after

RESPONSE: We agree that additional sample and data collection during the 12 weeks of TRE could be of great value; however, considering the primary aims of this project and an emphasis to keep subject burden low to increase feasibility, we are constrained to a pre-post assessment. If this pilot study shows TRE is safe and feasible in this population, the next step in clinical evaluation will be a larger scale and longer-term trial where additional timepoints for sample and data collection would certainly be pursued.

192 Are not weight stable…

RESPONSE: Thank you for bringing this to our attention so that we can make these criteria clearer. We have amended the text to now read, “Subjects ineligible to participate in this study are persons who: 9. Have a change in greater than 2 kg in body mass in the preceding three months.”

207 NfL abbreviation only introduced in line 308

RESPONSE: The text has been amended to introduce the NfL abbreviation at the first mention now found on line 216.

243 Only these values? Maybe include an extensive list of what will be measured

RESPONSE: The text has been amended to now read, “Upon returning to the clinic room, venous blood samples, specifically, a complete metabolic panel, complete blood count with differential, fasting lipid panel, and hemoglobin A1c will be obtained (Table 1).” We also added a new Table that includes an extensive list of each safety blood parameter being measured at the baseline and follow-up visit (Table 1).

247 What other questionnaires are included? Well-being etc. Include list with questionnaires and references

RESPONSE: We amended the text to expand on the description of the UHDRS clinical rating scale. We added a sentence that states, “Specifically, the assessment consists of a 15-item motor exam, the Symbol Digit Modalities Test, the Stroop Interference Test, a verbal fluency test, an interview-style assessment developed to rate frequency and severity of behavioral symptoms, and an evaluation of personal function which grades independence and capacity to complete daily tasks [27].” We also amended the text on line 245 to now state, “They will then be asked to complete a baseline demographics and health behaviors survey, which includes the Pittsburgh Sleep Quality Index, a dietary habits and physical activity questionnaire, and quality of life and personal health questionnaire (S3).” The exact survey and specific questionnaires can be found in the Supporting Information.

265 Were there clear instructions on zero-calorie beverages with sweeteners?

RESPONSE: Participants are specifically advised to only consume zero-calorie liquids including water, black coffee and/or tea during the fasting window. We specifically recommend against consuming zero-calorie beverages with artificial sweeteners, such as diet soda. We have expanded the text to include a statement on this. It now reads, “Beverages without calories, such as water, black coffee or tea, will be allowed; however, zero-calorie beverages with artificial sweeteners, such as diet soda, will not.”

281 Why only kcal and macronutrient composition? The quality of the food (e.g. highly processed) and the micronutrient composition are of great interest and should be considered as well.

RESPONSE: We agree the quality of food and micronutrient composition are of great interest. The SnapCalorieTM phone application collects participant meal entry data and sends an automated report to the study team that includes a timestamp of each meal entry, the photo taken by the participant, and a daily total of calories, fat, carbohydrates, protein, saturated fats, cholesterol, sodium, fiber, and sugar. Micronutrient estimations are beyond the capabilities of the app; however, we will examine the photos of meals to gain an understanding of the quality of food being consumed. We have amended the text to include this information. It now states, “The application uses artificial intelligence technology and manually entered nutrition information to analyze caloric composition which will be specifically used to estimate caloric content, and daily consumed fats, carbohydrates, proteins, saturated fats, cholesterol, sodium, fiber, and sugar [28]. Photos of meals will be used to describe the quality and types of foods being consumed.”

455 This is not the first intervention trial looking ate the effects of TRE in neurodegenerative diseases: Hansen B, Laczny CC, Aho VTE, et al Protocol for a multicentre cross-sectional, longitudinal ambulatory clinical trial in rheumatoid arthritis and Parkinson’s disease patients analysing the relation between the gut microbiome, fasting and immune status in Germany (ExpoBiome) BMJ Open 2023;13:e071380. doi: 10.1136/bmjopen-2022-07138

---

## [Decision Letter · Decision Letter 1]

30 Jan 2025

Time-restricted eating in early-stage Huntington’s disease: A 12-week interventional clinical trial protocol

PONE-D-24-40480R1

Dear Dr. Wells,

We’re pleased to inform you that your manuscript has been judged scientifically suitable for publication and will be formally accepted for publication once it meets all outstanding technical requirements.

Kind regards,

Prof. Jussi Sipilä, M.D., Ph.D., B.Soc.Sci.

Academic Editor

PLOS ONE

Reviewers' comments:

Reviewer's Responses to Questions

**Comments to the Author**

1. Does the manuscript provide a valid rationale for the proposed study, with clearly identified and justified research questions?

Reviewer #1: Yes

2. Is the protocol technically sound and planned in a manner that will lead to a meaningful outcome and allow testing the stated hypotheses?

Reviewer #1: Yes

3. Is the methodology feasible and described in sufficient detail to allow the work to be replicable?

Reviewer #1: Yes

4. Have the authors described where all data underlying the findings will be made available when the study is complete?

Reviewer #1: Yes

5. Is the manuscript presented in an intelligible fashion and written in standard English?

Reviewer #1: Yes

6. Review Comments to the Author

You may also provide optional suggestions and comments to authors that they might find helpful in planning their study.

Reviewer #1: The authors have applied all of my comments appropriately. I have no other comments to make. The article is ready to be accepted.

7. PLOS authors have the option to publish the peer review history of their article (what does this mean? ). If published, this will include your full peer review and any attached files.

**Do you want your identity to be public for this peer review?** For information about this choice, including consent withdrawal, please see our Privacy Policy .

Reviewer #1: **Yes: ** Dr Shah-Jalal Sarker

---

## [Editor Report · Acceptance letter]

PONE-D-24-40480R1

PLOS ONE

Dear Dr. Wells,

I'm pleased to inform you that your manuscript has been deemed suitable for publication in PLOS ONE. Congratulations! Your manuscript is now being handed over to our production team.

Kind regards,

on behalf of

Prof. Jussi Sipilä

Academic Editor

PLOS ONE